# CLUSTER-LEARNGENE: INHERITING ADAPTIVE CLUSTERS FOR SELF-ATTENTION

## ABSTRACT

In recent years, the merging of vast datasets with powerful computational resources has led to the emergence of large pre-trained models in the field of deep learning. However, the common practices often overgeneralize the applicability of these models, overlooking the task-specific resource constraints. To mitigate this issue, we propose **Cluster-Learngene**, which effectively condenses knowledge from an ancestry model and then initializes descendant models with varying scales of attention heads. Specifically, our method adaptively clusters attention heads of each layer in the ancestry model based on their density characteristics and extracts centroids of attention heads as the learngene. Moreover, we introduce a priority weight-sharing strategy that expands the learngene to initialize descendant models with varying scales of attention heads. Through extensive experimentation, we demonstrate that Cluster-Learngene is not only more efficient compared to other initialization methods but also customizes models with varying scales of attention heads according to downstream task resources.

## 1 INTRODUCTION

The evolution of deep learning has been profoundly influenced by the confluence of expansive data sources and robust computational capabilities. This collaboration has given rise to large pre-trained foundation models (Dosovitskiy et al., 2021; Devlin et al., 2019; Radford et al., 2021; Bubeck et al., 2023), particularly those built upon the Transformer architecture (Vaswani et al., 2017; Dosovitskiy et al., 2021), such as the Vision Transformers (ViTs) (Dosovitskiy et al., 2021). The pre-trained foundation models, being widely deployed in various devices like smartphones or edge devices, serve as the initialization point (Hanin & Rolnick, 2018; Arpit et al., 2019; He et al., 2016; Zhang et al., 2021; Wang et al., 2022; 2023) for diverse downstream applications. However, this dominant methodology implicitly assumes that a one-size-fits-all approach, *i.e.*, the entirety of the foundational model is universally apt for every application, neglecting the specific resource constraints (*e.g.*, memory, FLOPs, or latency) inherent to certain downstream tasks. Such an assumption can be impractical in myriad practical scenarios, especially when deploying models on resource-limited devices. Furthermore, not all tasks demand the full power of these extensive foundation models. This naturally raises a pivotal question: *Can we extract and harness the condensed part of these foundation models to achieve a harmonious balance between accuracy and resource efficiency?*

To achieve the goal of efficiently initializing models, (Wang et al., 2022; 2023) introduce the innovative *Learngene* framework inspired by the observation of genes (cf. Fig. 1 (a)). As showcased in Fig. 1 (b), *Learngene* framework is designed in two pivotal stages. In the first stage, the significant knowledge is condensed from a large **ancestry model** into a more compact part termed as **learngene**. In the next stage, this learngene is inherited to initialize the **descendant models** of assorted scales. [1] Previous works (Wang et al., 2022; 2023) predominantly focus on extracting a few integral layers as the learngene and manually stacking them with the randomly initialized layers.

However, such approaches struggle with inherent limitations: (i) The strategy of extracting certain integral layers overlooks the potential existence of learngene within these layers, leading to the preservation of many redundant weights. (ii) The approach of manually stacking the learngene with

---

[1]The terms "foundation model" and "ancestry model," as well as "downstream model" and "descendant model," are interchangeably utilized unless distinctions are explicitly mentioned.

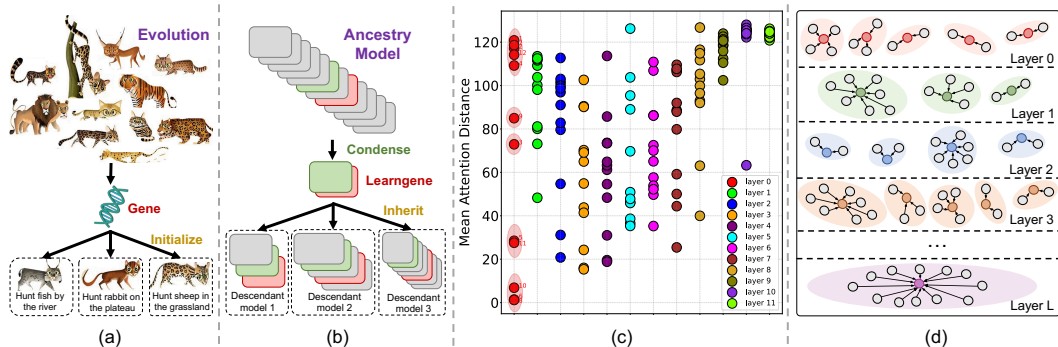

Figure 1: (a) The ancestry of biological organisms condenses evolutionary information into information-dense genes to initialize their diverse descendants (Zador, 2019; Hasson et al., 2020). (b) The *Learngene* framework condenses the significant knowledge from an ancestry model into a more compact part termed learngene and then inherited to initialize the descendant models of assorted scales. (c) The distribution density of attention heads across the different layers of the ancestry model, which employs the DeiT-B (Touvron et al., 2021). (d) An illustration of our idea.

randomly initialized layers lacks the adaptability to scale the model, preventing the initialization of downstream models with custom dimensions.

As mentioned earlier, the *Learngene* framework aims to preserve the most generalizable part of the ancestry model while eliminating redundant weights that weaken representational capacity. Recent studies (Raghu et al., 2021; Xie et al., 2023) have visualized the mean attention distance of ViTs, offering deeper insights into weight redundancy among attention heads across different layers. As illustrated in Fig. 1 (c), the lower layers focus on both local and global perspectives, leading to a more sparse density of attention heads. Conversely, the higher layers prioritize a global context, resulting in a compact density. A notable observation is the repetitive functionality across many attention heads especially in the higher layers, which inevitably leads to weight redundancy.

Inspired by the above observation, we propose the Cluster-Learngene, an innovative approach that adaptively extracts the cluster centroids of the attention heads (*i.e.*, **head centroids**) across each layer of the ancestry model as learngene. To extract them, we cluster the attention heads within each layer of the ancestry model based on their density characteristics. As depicted in Fig. 1 (c-d), the attention heads in the first layer exhibit a sparse density, resulting in five clusters, whereas the attention heads in the last layer cluster more compactly, forming a single group. Our Cluster-Learngene preserves the critical parameters containing significant knowledge because the extracted head centroids represent attention heads with similar semantics.

In the inheriting stage, to expand the learngene into various descendant models, we adopt the priority weight-sharing. We start by ranking the head centroids based on the size of their respective clusters, arranging them in descending order of priority. Subsequently, we perform weight-sharing by distributing these head centroids to initialize the attention heads of the descendant models. If the number of attention heads in a specific layer aligns perfectly with the number of centroids, they are evenly shared. However, if they fail to align perfectly, any remaining centroids are shared according to the remainder.

Our **contributions** can be summarized as follows: (i) We propose the adaptive clustering of attention heads to extract head centroids as the learngene, ensuring the preservation of significant knowledge within the ancestry model. (ii) To achieve the initialization of descendant models with **varying scales of attention heads**, we introduce priority weight-sharing that favors head centroids within larger clusters. (iii) Comprehensive experimental evaluations across datasets of different scales reveal that Cluster-Learngene not only outperforms traditional initialization strategies but also stands toe-to-toe with more resource-demanding fine-tuning methodologies.

## 2 METHODOLOGY

*Learngene* framework is primarily divided into two phases in Fig. 1 (b): the significant knowledge is condensed from an ancestry model into a more compact part termed as learngene and then inherited

to initialize the descendant models of assorted scales. Specifically, in phase 1, our Cluster-Learngene selects mean attention distance as the density metric and uses it to cluster the head centroids of each layer in the ancestry model as the **learngene**, because these head centroids can effectively represent attention heads with similar semantics. The pseudocode for this phase is presented in Algorithm 1. In phase 2, Fig. 2 illustrates priority weight-sharing for initializing attention heads in descendant models. Next, we briefly introduce some preliminaries related to ViTs.

## 2.1 PRELIMINARY

In the ViT architecture, an input image is first divided into $N$ non-overlapping patches, and each patch is linearly embedded into a flat vector of size $D$. The ViT encoder consists of alternating layers of multi-head self-attention (MSA) and position-wise feed-forward network (FFN) blocks. Let $H$ denote the total number of heads in each layer. For the $h^{th}$ head, the query $\mathbf{Q}_h \in \mathbb{R}^{N \times d_k}$, key $\mathbf{K}_h \in \mathbb{R}^{N \times d_k}$, and value $\mathbf{V}_h \in \mathbb{R}^{N \times d_v}$ are linearly generated through learned weight matrices $\mathbf{W}_h^Q \in \mathbb{R}^{D \times d_k}$, $\mathbf{W}_h^K \in \mathbb{R}^{D \times d_k}$, and $\mathbf{W}_h^V \in \mathbb{R}^{D \times d_v}$, where $d_k$ and $d_v$ are the dimensions of the key and value vectors, respectively. The SA mechanism of the $i$-th head can be represented as:

$$\mathbf{A}^h = \text{Attention}(\mathbf{Q}_h, \mathbf{K}_h, \mathbf{V}_h) = \text{softmax}\left(\frac{\mathbf{Q}_h \mathbf{K}_h^\top}{\sqrt{d_k}}\right) \mathbf{V}_h. \tag{1}$$

MSA allows the model to jointly attend to information at different positions from different representational subspaces at different positions:

$$\text{MultiHead}(\mathbf{Q}, \mathbf{K}, \mathbf{V}) = \text{Concat}(\mathbf{A}^1, \dots, \mathbf{A}^H)\mathbf{W}^O, \tag{2}$$

where $\mathbf{W}^O \in \mathbb{R}^{Hd_v \times D}$ is a learned weight matrix. Besides, the FFN can be formulated as:

$$\text{FFN}(\mathbf{x}) = \text{ReLU}(\mathbf{x}\mathbf{W}_1 + \mathbf{b}_1)\mathbf{W}_2 + \mathbf{b}_2, \tag{3}$$

where $\mathbf{x} \in \mathbb{R}^{N \times D}$ is the input, $\mathbf{W}_1 \in \mathbb{R}^{D \times d_{ff}}$ and $\mathbf{W}_2 \in \mathbb{R}^{d_{ff} \times D}$ are the weight matrices, and $\mathbf{b}_1 \in \mathbb{R}^{d_{ff}}$ and $\mathbf{b}_2 \in \mathbb{R}^D$ are the bias vectors. $d_{ff}$ is the dimension of the intermediate layer.

## 2.2 ADAPTIVELY LEARNGENE CLUSTERING

**Density metric on attention heads.** Given a pre-trained ancestry model with $L$ layers and $H_a$ attention heads per layer, let the attention weights for the $h^{th}$ head in the $l^{th}$ layer be denoted by the matrix $\mathbf{A}^{(l,h)} \in \mathbb{R}^{N \times N}$. The element $A_{i,j}^{(l,h)}$ represents the attention weight from position $i$ to position $j$. The distance between any two positions $i$ and $j$ in the sequence can be straightforwardly defined as $|i - j|$. Consequently, the distance matrix $\mathbf{T} \in \mathbb{R}^{D \times D}$ can be described with $T_{i,j} = |i - j|$. The mean attention distance for the $h^{th}$ head in the $l^{th}$ layer, encapsulating the weighted distance for each position $i$ across the sequence, is given by:

$$MeanDist^{(l,h)} = \frac{1}{D} \sum_{i=1}^{D} \sum_{j=1}^{D} A_{i,j}^{(l,h)} \times T_{i,j}. \tag{4}$$

To deduce this metric for every head across all layers, iterate the above computation for every $l \in \{1, \dots, L\}$ and $h \in \{1, \dots, H_a\}$. As depicted in Fig. 1 and Appendix A, while the lower layers simultaneously attend to both local and global features, leading to a more dispersed distribution of attention heads, the higher layers predominantly focus on global aspects, causing a tighter concentration of attention heads. As a result, there is a significant overlap in the semantic representations among many attention heads, especially in the higher layers, leading to weight redundancy.

**Adaptively clustering.** Motivated by the empirical observations, we extract cluster centroids (Schubert et al., 2017; Bushra & Yi, 2021; Bhattacharjee & Mitra, 2021) of attention heads in ViTs as the learngene inherited into the descendant models, thus aggregating similar semantics into the head centroids. To realize this, we select $MeanDist$ as a density metric for adaptively clustering the attention heads of the ancestry model at each layer, without setting the number of clusters in advance. This realization prompts the formulation of the definitions and lemmas, which scaffold our adaptive clustering approach.

---

**Algorithm 1:** Pseudocode of Adaptively Learngene Clustering

---

1 **Input:** Number of layers in ViT as $L$, set of attention heads in the $l^{th}$ layer as $S_l$, radius as $Eps$, density threshold as $MinHds$, and distance function as $Dist$.
2 **Output:** The centroids of attention head in all clusters.
3 Initialize all attention heads as unvisited and an empty list for clusters
4 **for** $l = 1, \ldots, L$ **do**
5      **foreach** *attention head $a$ in $S_l$* **do**
          // Iterate set of attention heads in the $l^{th}$ layer
6          **if** *$a$ is not visited* **then**
7             Mark $a$ as visited, $NeighborHds \leftarrow$ all attention heads within $Eps$ distance of $a$
            // Initialize neighbors
8          **end**
9          **if** *number of $NeighborHds \geq MinHds$* **then**
10             $C \leftarrow$ new cluster, Add $a$ to cluster $C$        // Start a new cluster
11             **foreach** *attention head $b$ in $NeighborHds$* **do**
              // Expand neighborhood
12               **if** *$b$ is not visited* **then**
13                 Mark $b$ as visited
14                 $NeighborHds' \leftarrow$ all attention heads within $Eps$ distance of $b$
15               **end**
16               **if** *number of $NeighborHds' \geq MinHds$* **then**
17                 $NeighborHds = NeighborHds \cup NeighborHds'$
18               **end**
19               **if** *$b$ is not yet a member of any cluster* **then** Add $b$ to cluster $C$
20             **end**
21             Add $C$ to the list of clusters        // Consolidate clusters
22          **end**
23          **else** Mark $a$ as noise
24      **end**
25 **end**

---

**Definition 1 (Eps-neighborhood of an attention head).** The Eps-neighborhood of an attention head $a$, denoted as $N_{Eps}(a)$, is defined as: $N_{Eps}(a) = \{b \in S \mid Dist(a, b) \leq Eps\}$, where $Dist(a, b)$ denotes the difference in $MeanDist$ values between attention heads $a$ and $b$. Our approach could require for each head in a cluster that there are at least a **Min**imum number of **H**eads ($MinHds$) in an Eps-neighborhood of that head.

**Definition 2 (density-reachable).** Transitioning from the neighborhood concept, an attention head $a$ is considered density-reachable from another head $b$ with respect to $Eps$ and $MinHds$ if there is a sequence of heads $a_1, \ldots, a_n$ such that $a_1 = b, a_n = a$, and each head in this sequence lies within the Eps-neighborhood of its preceding head.

**Definition 3 (density-connected).** Broadening our purview, attention heads $a$ and $b$ are labeled density-connected with respect to $Eps$ and $MinHds$ if there exists an intermediary head $o$ from which both $a$ and $b$ are density-reachable.

Considering all attention heads in layer $l$ as $S_l$, a cluster $C$ based on $Eps$ and $MinHds$ is identified as a non-empty subset of $S_l$ that satisfies the conditions: (i) **Maximality:** For any heads $a$ and $b$ in the sequence, if $a$ resides within $C$ and $b$ is **density-reachable** from $a$ dictated by $Eps$ and $MinHds$, then $b$ seamlessly becomes part of $C$. (ii) **Connectivity**: Within $C$, each pairing $a, b$ maintains a **density-connection**, anchored by $Eps$ and $MinHds$. Therefore, upon satisfying these two conditions, we select all attention heads centrally positioned within the clusters as the learngene, which is then inherited into the descendant models. The pseudo-code is summarized in Algorithm 1. The lemma presented below is pivotal in substantiating the correctness of our clustering algorithm.

**Lemma 1.** Presuming an attention head $a$ belongs to $S_l$ and satisfies the condition $|N_{Eps}(a)| \geq MinHds$. Then, the set $O = \{o \mid o \in S_l$ and $o$ is density-reachable from $a$ with respect to $Eps$ and $MinHds\}$ collectively shapes a cluster.

Figure 2: Illustration of priority weight-sharing. The darker the color, the larger the cluster size associated with the head centroid.

## 2.3 LEARNGENE INHERITING

**Expanding self-attention clusters with priority weight-sharing.** Building on the aforementioned method, we extract the head centroids as the learngene. For an ancestry model with $L$ layers, $L$ layers of head centroids are extracted. The $l^{th}$ layer has $c_l$ head centroids of weight $\mathbf{A}^{(l,1)}, \ldots, \mathbf{A}^{(l,c_l)}$. Importantly, the head centroids at each layer are sorted in descending order based on the size of their respective cluster, *i.e.*, centroids representing more attention heads in the ancestry model are ranked higher. These head centroids condense significant knowledge and ensure the initialization of descendant models without performance degradation. Assume the descendant model has $H_d$ attention heads for each layer. To achieve the desired expansion of heads to initialize the descendant models, we adopt the priority weight-sharing and Fig. 2 illustrates two scenarios:

- When $H_d$ is divisible by $c_l$: The weights of head centroids are shared $\frac{H_d}{c_l}$ times in sequence. For instance, centroids of weights $\mathbf{A}^{(L,1)}$ and $\mathbf{A}^{(L,2)}$ each share their weights across four attention heads, which are then directly assigned to eight attention heads of the descendant model in layer $L$.

- When $H_d$ is not divisible by $c_l$: The weights of the head centroids are sequentially shared $\left\lfloor \frac{H_d}{c_l} \right\rfloor$ times, followed by appending $\mathbf{A}^{(l,1)}, \ldots, \mathbf{A}^{(l,H_d \bmod c_l)}$ at the end. As an illustration, we share the centroids of weights $\mathbf{A}^{(1,1)}, \ldots, \mathbf{A}^{(1,5)}$ once and then append $\mathbf{A}^{(1,1)}, \ldots, \mathbf{A}^{(1,3)}$, thus initializing eight attention heads of the descendant model in the first layer.

According to the adjustments in the number of attention heads, the weights $\mathbf{W}^O$ of the projection layer are also proportionally pruned and then inherited by the descendant models. [2]

**Model Variant.** For the weights of FFN in the descendant models, we adopt direct inheriting from the ancestry model or random initialization, and the results are discussed in Experiment 3.2. For the attention heads in the descendant models, we introduce the hyperparameter $\omega = \frac{H_a}{H_d}$ to denote the factor by which the number of attention heads is reduced compared to the ancestry model. In addition to uniformly setting the number of attention heads for each layer with the hyperparameter $\omega$, we also explore two other possibilities in Experiment 3.3: incrementing and decrementing the count of attention heads with layer depth.

**Complexity Analysis.** Comparing the model complexities of all attention heads in the ancestry model, our approach reduces the model complexity of the initialized attention heads in descendant models by $\frac{LH_a}{\sum_1^L c_l}$. Detailed derivations are provided in the Appendix E.

## 3 EXPERIMENTS

### 3.1 EXPERIMENTAL SETTING

**Datasets.** To condense the learngene, we employ the ImageNet-1K, a collection of 1.2 million training images and 50,000 validation images distributed across 1,000 classes as part of the ILSVRC2012 competition Deng et al. (2009). After initializing the descendant models with the Learngene, we proceed to fine-tune these models on diverse downstream tasks. These tasks include Tiny-ImageNet Le & Yang (2015), Food-101 Bossard et al. (2014), CUB-200 Wah et al. (2011), CIFAR-10 Krizhevsky et al. (2009), CIFAR-100 Krizhevsky et al. (2009), and iNaturalist-2019 Tan et al. (2019). [3]

---

[2]Please see Appendix D for more details.

[3]Please refer to Appendix B for detailed dataset descriptions.

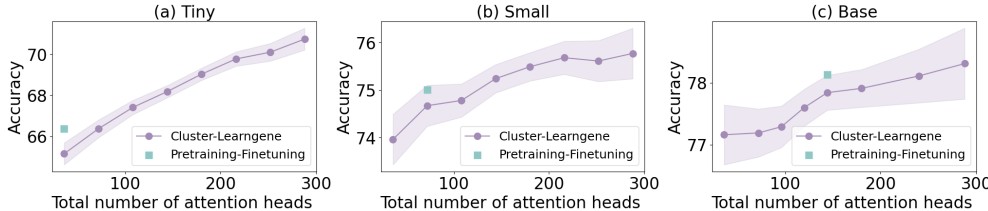

Figure 3: **Initializing descendant models with varying scales of attention heads.** We fine-tune 50 epochs for all models. In (a), the hyperparameter $\omega$ takes values ranging from a maximum of 1 to a minimum of $\frac{1}{8}$ (i.e., the number of attention heads in descendant models is eight times that of the ancestry model). In (b), $\omega$ ranges from a maximum of 2 to a minimum of $\frac{1}{4}$. Continuing this pattern, in (c), $\omega$ ranges from a maximum of 4 to a minimum of $\frac{1}{2}$.

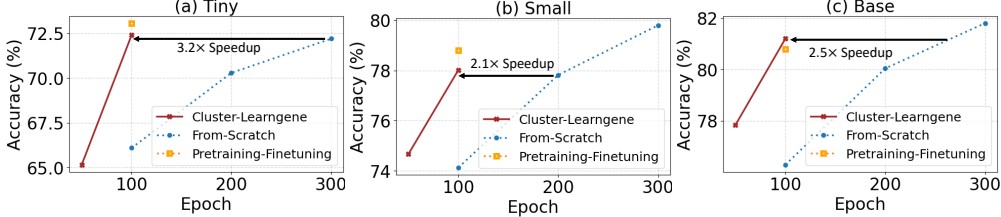

Figure 4: **Faster convergence.** Different points represent results for varying epochs and the hyperparameter $\omega$ is set to 1.0 for our method.

**Baselines.** Both the ancestry model and descendant models are variants derived from DeiT (Touvron et al., 2021). In terms of width, there are three types of DeiT: **Tiny**, **Small**, and **Base**. For more training details and hyperparameters, see Appendix C. We conduct a comparative analysis of our approach for initializing descendant or downstream models, as follows: (i) Pretraining-Finetuning: This approach pre-trains DeiT on ImageNet and subsequently fine-tunes the entire model on downstream tasks. (ii) From-Scratch: We commence with a randomly initialized DeiT model and exclusively train it on the downstream datasets. (iii) Heuristic-Learngene (Wang et al., 2022): This strategy involves extracting the last three layers from a DeiT model pre-trained on ImageNet. These layers are then stacked with randomly initialized lower layers to construct a new model. (iv) Weight-Transformation (Zhang et al., 2022a): This method employs Weight Transformation to pre-train DeiT on ImageNet, followed by fine-tuning the entire model to adapt it to specific downstream tasks. (v) Auto-Learngene (Wang et al., 2023): The first six layers are extracted from the DeiT and then stacked with randomly initialized higher layers to initialize the descendant models.

## 3.2 MAIN RESULTS OF MODEL INITIALIZATION

In this section, we validate the capabilities of Cluster-Learngene in efficiently initializing models and measure model performance with Top-1 accuracy.

**Initializing descendant models with varying scales of attention heads.** We expand the varying number of attention heads to initialize the descendant models by adjusting the hyperparameter $\omega$, catering to downstream resource constraints. As illustrated in Fig. 3, in the case of Tiny-scale descendant models, when the total number of attention heads is as low as 32, the performance of Cluster-Learngene is slightly below that of Pretraining-Finetuning. However, as the total number of attention heads increases, Cluster-Learngene surpasses Pretraining-Finetuning. Particularly noteworthy is the improvement of over **3%** when there are 288 attention heads because a sufficient number of attention heads are initialized by the learngene, which holds significant knowledge. Therefore, our method resolves the limitations of the one-size-fits-all approach seen in Pretraining-Finetuning.

**Faster convergence.** We provide a detailed comparison of training efficiency between our approach and From Scratch. As shown in Fig. 4, Cluster-Learngene requires only **3.2** $\times$ less training overhead compared to From Scratch on Tiny-scale descendant models. A key advantage of our approach is that descendant models initialized with the learngene achieve faster convergence, owing to a superior initialization point.

Table 1: **Initialization of descendant models with diverse training samples.** The symbol ↑ denotes the performance gap between our approach and the From-Scratch method. Cluster-Learngene initializes the descendant model over 50 training epochs. In contrast, From-Scratch results are achieved after 300 training epochs.

| Training data | From-Scratch | Cluster-Learngene |
|---|---|---|
| 100% IN-1K | 81.80 | 78.65 |
| 50% IN-1K | 74.70 | 76.44(↑**1.74**) |
| 25% IN-1K | 65.73 | 75.97(↑**10.24**) |

Table 2: **Increment or decrement the count of attention heads.** "Decrementing" denotes halving the number of attention heads in the first four layers, reducing them by a quarter in the middle four layers, and maintaining them in the last four layers relative to the ancestry model. Conversely, "Incrementing" represents the opposite pattern.

| Model | Decrementing | Incrementing |
|---|---|---|
| Tiny | 76.56 | **78.01** |
| Small | 79.47 | **81.29** |
| Base | 80.18 | **81.65** |

**Higher data efficiency.** We further conduct experiments on Base-scale descendant models over different percentages of training data from ImageNet-1K (IN-1K). As shown in Tab. 1, while our method does not outperform the From-Scratch on the entire dataset, its performance exhibits greater stability as the amount of training data decreases. For instance, with only 25% of the training data, Cluster-Learngene outperforms From-Scratch by **10.24%**. This higher data efficiency of our method is attributed to the significant knowledge within the learngene, which helps descendant models mitigate overfitting, especially in scenarios with limited data.

**Efficiently initializing large models on ImageNet.** The experimental results in Tab. 3 highlight several advantages of our approach: (i) When compared to classical initialization methods, our approach exhibits superior performance. For example, on Base-scale descendant models, From-Scratch achieves an accuracy of 69.88%, whereas Cluster-Learngene achieves 77.84%. Furthermore, Cluster-Learngene maintains comparable performance with Pretraining-Finetuning while reducing inherited parameters by **66%** and the inherited count of attention heads by **70.1%**. Cluster-Learngene* even outperforms Pretraining-Finetuning by 3.34% with fewer parameters. (ii) Our method inherits fewer parameters than weight compression methods like Weight-Transformation. On Base-scale descendant models, Weight-Transformation inherits 44.0 million parameters, whereas our method inherits only 29.1 million parameters. (iii) In comparison to other *Learngene* methods, our approach efficiently initializes descendant models with varying scales of attention heads. For instance, on Base-scale descendant models, while Auto-Learngene inherits 42.4 million parameters, Cluster-Learngene inherits only 29.1 million parameters. Moreover, Cluster-Learngene* outperforms Auto-Learngene with an accuracy of **81.47%** compared to 78.04%. This improvement is attributed to the consideration of redundancy in initializing the internal attention heads of ViT, thus diversifying the representational capacity of attention heads.

## 3.3 Analysis and Ablation

In this section, we provide further analysis and ablation of Cluster-Learngene. Unless otherwise specified, we conduct experiments on **CIFAR-100** and use **Small-scale DeiT** as the ancestry model.

**Variation in the count of attention head with model depth.** Tab. 2 presents two scenarios where the number of attention heads varies across different layers. Across all descendant model configurations, "Incrementing" consistently outperforms "Decrementing" by a margin of 1.45% in terms of accuracy. These findings align with previous research (Michel et al., 2019; Liu et al., 2021), which suggests that setting more attention heads in higher layers can assist these layers in learning more abstract and high-level feature representations.

**Qualitative visualization.** We visualize attention representations to explain which significant knowledge is inherited by learngene, as shown in Fig. 5. To reduce non-linear effects and enhance the saliency of the display, we set the power exponent to $\gamma = 0.25$. Head centroids from head 1, 2, 4, and 5 of the first layer in the ancestry model are clustered to initialize head 1 in the descendant model. Similarly, head centroids from head 3 of the first layer in the ancestry model are used to initialize head 2 in the descendant model, and so on. Then, weight-sharing is applied to expand head centroids, *e.g.*, sharing twice to initialize the descendant model.

Table 3: **Trade-off between efficiency and accuracy on ImageNet-1K.** The terms "T-Params" and "I-Params" denote the **T**otal number of parameters and the number of **I**nherited Parameters in the downstream/descendant models, respectively. Similarly, "T-Head" and "I-head" refers to the **T**otal number of attention heads and the **I**nherited count of attention heads in the downstream/descendant models. **Cluster-Learngene** and **Cluster-Learngene**$^*$ respectively represent random initialization and direct inheriting of FFN. The underline denotes our method compared to From-Scratch. The symbols ↓ and ↑ represent the difference between our method and Pretraining-Finetuning. To verify the capability of SoLe to provide rapid initialization for the model, we designate a 50-epoch training period for fine-tuning all models.

| Model | Method | T-Head | I-Head | T-Params (M) | I-Params (M) | FLOPs (G) | Acc (%) |
|---|---|---|---|---|---|---|---|
| Tiny | Pretraining-Finetuning | 36 | 36 | 5.5 | 5.5 | 1.1 | 66.36 |
| | From-Scratch | 36 | 0 | 5.5 | 0 | 1.1 | 58.93 |
| | Heuristic-Learngene | 36 | 9 | 5.5 | 1.3 | 1.1 | 62.45 |
| | Weight-Transformation | 36 | 18 | 3 | 2.9 | 1.2 | 65.08 |
| | Auto-Learngene | 36 | 18 | 5.5 | 2.6 | 1.1 | 65.34 |
| | **Cluster-Learngene** | 36 | 27(↓**25.0%**) | 4.2 | 2(↓**63.6%**) | 0.8(↓**27.3%**) | 65.15 |
| | **Cluster-Learngene**$^*$ | 36 | 27 | 4.2 | 4.2 | 0.8 | 71.68(↑**5.32**) |
| Small | Pretraining-Finetuning | 72 | 72 | 21.6 | 21.6 | 4.3 | 75.01 |
| | From-Scratch | 72 | 0 | 21.6 | 0 | 4.3 | 68.41 |
| | Heuristic-Learngene | 72 | 18 | 21.6 | 5.3 | 4.3 | 72.39 |
| | Weight-Transformation | 72 | 36 | 11 | 11 | 4.3 | 75.45 |
| | Auto-Learngene | 72 | 36 | 21.6 | 10.6 | 4.3 | 75.90 |
| | **Cluster-Learngene** | 72 | 30(↓**58.3%**) | 16.3 | 7.5(↓**65.3%**) | 3.2(↓**25.6%**) | 74.67 |
| | **Cluster-Learngene**$^*$ | 72 | 30 | 16.3 | 16.3 | 3.2 | 78.72(↑**3.71**) |
| Base | Pretraining-Finetuning | 144 | 144 | 85.6 | 85.6 | 16.9 | 78.13 |
| | From-Scratch | 144 | 0 | 85.6 | 0 | 16.9 | 69.88 |
| | Heuristic-Learngene | 144 | 36 | 85.6 | 21.2 | 16.9 | 75.83 |
| | Weight-Transformation | 144 | 72 | 44.0 | 44.0 | 17.0 | 78.76 |
| | Auto-Learngene | 144 | 72 | 85.6 | 42.4 | 16.9 | 78.04 |
| | **Cluster-Learngene** | 144 | 43(↓**70.1%**) | 64.4 | 29.1(↓**66.0%**) | 12.7(↓**24.9%**) | 77.84 |
| | **Cluster-Learngene**$^*$ | 144 | 43 | 64.4 | 64.4 | 12.7 | 81.47(↑**3.34**) |

In the first layer, heads 1, 2, 4, and 5 form the largest cluster, showcasing a predominant concentration of attention representations along the main diagonal. This representation pattern repeats across multiple heads. Moreover, the first layer exhibits a diverse range of learned semantics, containing two other less frequent representation patterns, *i.e.*, the patterns head 3 and head 4. Notably, head 4 captures more abstract and high-level representations, as its attention distribution resembles that of the final layer. Consequently, the first layer of the learngene captures three critical representation patterns from the ancestry model, inheriting them into the descendant models. In contrast, the representations in the final layer of the ancestry model exhibit significant repetition, leading to the clustering of a single-head centroid for initializing the attention heads of the descendant model.

**Transfer learning results for the descendant/downstream models.** Tab. 4 illustrates the results of transfer learning for descendant models trained on various downstream tasks. Our Cluster-Learngene significantly outperforms both From-Scratch and Weight-Transformation. When compared to other Learngene methods, such as Auto-Learngene, we observe substantial improvements. Notably, on the Tiny-ImageNet (Tiny-IN) and iNaturalist-2019 (iNat-2019) datasets, Cluster-Learngene outperforms Auto-Learngene by **16.75%** and **6.89%**, respectively. These results highlight the superior capability of Cluster-Learngene in efficiently initializing descendant models.

Furthermore, on most datasets, the performance of Cluster-Learngene closely matches that of Pretraining-Finetuning, where the entire model is fine-tuned. Interestingly, on Tiny-ImageNet, Cluster-Learngene exceeds Pretraining-Finetuning by **11.65%** in accuracy. This phenomenon can be attributed to the more universally significant knowledge within learngene, allowing it to adapt effectively to various downstream tasks. In contrast, Pretraining-Finetuning, due to its reuse of the entire model, may lead to negative transfer (Wang et al., 2019; Zhang et al., 2022b) effects from unfavorable parts of the model in downstream tasks.

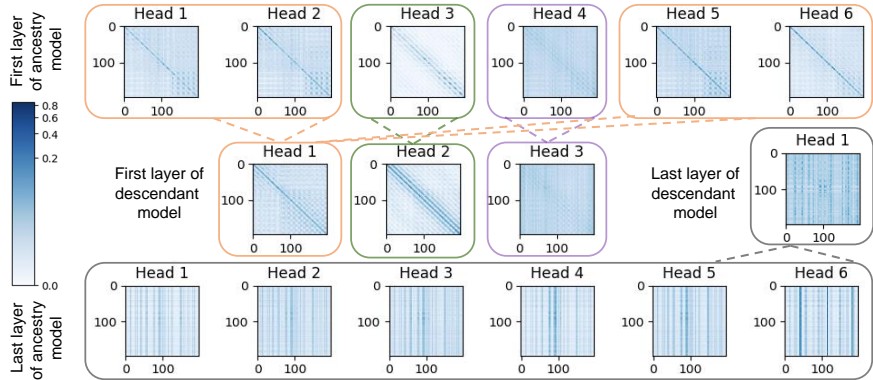

Figure 5: **Visualization of attention representations** $(197 \times 197)$. We perform the following normalization operation on all attention heads $\mathbf{A}$ of the ancestry model and descendant model: $\left( \frac{\mathbf{A}_{i,j}}{255} \right)^{\gamma}$. The descendant model is trained for 50 epochs, and $\omega$ is set to $\frac{1}{4}$.

Table 4: **DeiT-Small Results on downstream datasets.** "I-Params" means the number of **I**nherited parameters in the downstream/descendant models, measured in MB. $\uparrow$ represents the performance improvement achieved by Cluster-Learngene, when compared to the best method excluding Pretraining-Finetuning. All results are derived from the 6-layer downstream models.

| Method | I-Params | Tiny-IN | Food-101 | CUB-200 | CIFAR-10 | CIFAR-100 | iNat-2019 |
|---|---|---|---|---|---|---|---|
| Pretraining-Finetuning | 10.5 | 72.24 | 87.8 | 78.13 | 97.59 | 84.43 | 68.48 |
| From-Scratch | 0 | 61.24 | 74.64 | 62.75 | 92.49 | 73.32 | 50.79 |
| Heuristic-Learngene | 5.6 | 62.37 | 77.09 | 72.64 | 93.12 | 78.13 | 53.21 |
| Weight-Transformation | 10.5 | 64.56 | 81.79 | 70.28 | 93.67 | 75.98 | 59.83 |
| Auto-Learngene | 10.5 | 67.14 | 80.25 | 73.31 | 93.58 | 79.49 | 59.92 |
| **Cluster-Learngene** | 7.5 | 83.89(↑**16.75**) | 87.05(↑**5.26**) | 76.84(↑**3.53**) | 96.90(↑**3.23**) | 83.55(↑**4.06**) | 66.81(↑**6.89**) |

## 4 RELATED WORK

**Model Initialization:** Over the years, various initialization techniques have been proposed including the popular random initialization, Xavier initialization (Glorot & Bengio, 2010) and the Kaiming initialization (He et al., 2016). Recently, the use of pre-trained foundation models has gained prominence as an initialization strategy before fine-tuning for specific tasks (Dosovitskiy et al., 2021; Devlin et al., 2019; Radford et al., 2021; Yang et al., 2022; Ni et al., 2022; Bubeck et al., 2023). However, such an approach necessitates pre-training separate models for each downstream task, which can lead to substantial computational resource consumption. In contrast, Cluster-Learngene presents a unique model initialization method that alleviates the need for multiple pre-training steps.

**Density-based Clustering:** Clustering aims to group similar data points together while separating dissimilar ones. A wide array of approaches has been explored, including partitioning-based clustering (Hamerly & Elkan, 2003; Ahmed et al., 2020), hierarchical clustering (Murtagh & Contreras, 2012; Cohen-Addad et al., 2019), and density-based clustering (Kriegel et al., 2011; Schubert et al., 2017; Bushra & Yi, 2021; Bhattacharjee & Mitra, 2021), and so on. In particular, density-based clustering operates by taking into account the density and distance relationships between data points to form clusters. Inspired by this, our method adopts a similar principle by assessing the density of attention heads to retain essential head centroids that represent significant knowledge.

## 5 CONCLUSION

In this paper, we propose Cluster-Learngene, a novel approach that involves the adaptive clustering of attention heads to extract head centroids as the learngene. Subsequently, we adopt the priority weight-sharing to expand the learngene for initializing descendant models with varying scales of attention heads, enabling adaptation to diverse downstream resource constraints. Extensive experiments validate the efficiency and scalability of our initialization method.

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

## APPENDIX

## A  MEAN ATTENTION DISTANCE IN DEIT-S AND DEIT-TI

Fig. 6 illustrates the mean attention distance for two other variants of DeiT. In both variants, the lower layers exhibit a dual focus on both local and global aspects, resulting in a relatively sparse distribution of attention heads. Conversely, the higher layers prioritize the global context, leading to a more compact distribution of attention heads. Importantly, many attention heads in these layers exhibit repetitive functionality, contributing to weight redundancy.

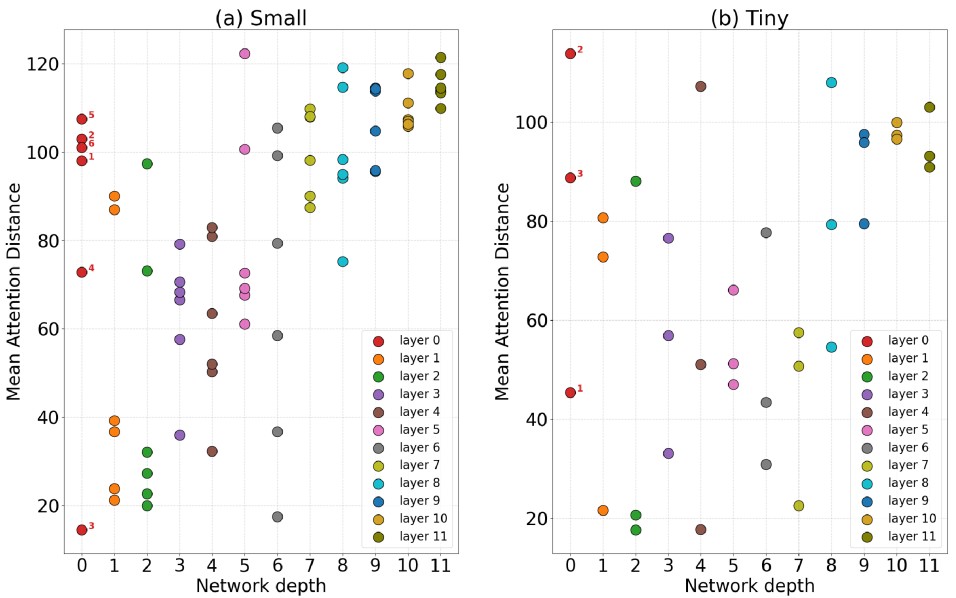

Figure 6: The distribution density of attention heads across the different layers of the ancestry model, which employs the DeiT-S and DeiT-Ti (Touvron et al., 2021).

## B  DOWNSTREAM DATASETS

Tab. 5 presents the details of all downstream tasks.

## C  TRAINING SETTINGS

During the learngene clustering, We set $Eps = 10, MinHds = 1$. In the learngene inheriting phase, we train the descendant models on downstream tasks for 500 epochs, including a 10-epoch warm-up

Table 5: Characteristics of the downstream datasets

| Dataset | # Total | #Training | #Validation | #Testing | #Classes |
|---|---|---|---|---|---|
| CUB-200-2011 (Wah et al., 2011) | 11,788 | 5,394 | 600 | 5,794 | 200 |
| CIFAR10 (Krizhevsky et al., 2009) | 65,000 | 50,000 | 5,000 | 10,000 | 10 |
| CIFAR100 (Krizhevsky et al., 2009) | 65,000 | 50,000 | 5,000 | 10,000 | 100 |
| Food101 (Bossard et al., 2014) | 101,000 | 75,750 | 25,250 | 0 | 101 |
| Tiny-ImageNet (Le & Yang, 2015) | 120,000 | 100,000 | 10,000 | 10,000 | 200 |
| iNat-2019 (Tan et al., 2019) | 268,243 | / | / | / | 1010 |

period, except for iNaturalist-2019, where we train for 100 epochs with a 5-epoch warm-up. The initial learning rate is set to $5 \times 10^{-4}$ for most tasks, except for Stanford Cars where it is $5 \times 10^{-3}$, and a weight decay of 0.05. All models are implemented in PyTorch Paszke et al. (2019) and trained on NVIDIA RTX 3090 GPUs.

## D  PROJECTION LAYER

According to the adjustments in the number of attention heads, the weights $\mathbf{W}^O$ of the projection layer are also proportionally pruned or expanded with the hyperparameter $\omega$ and then inherited by the descendant models. Additionally, we directly inherit the weights of layer normalization, patch embeddings, and position embeddings in the ancestry model, which constitute only a small fraction of all weights.

## E  COMPLEXITY ANALYSIS

For an ancestry model with a total of $L$ layers, each containing $H_a$ attention heads, the total parameters of its attention heads amount to $LH_a(2d_k + d_v)D$. Cluster-Learngene condenses each layer of the ancestry model into $c_l$ head centroids. When all these head centroids are inherited by descendant models through priority weight-sharing, the total parameters of the attention heads in the descendant models become $\sum_1^L c_l(2d_k + d_v)D$. Therefore, the relative reduction in model complexity of the descendant models' attention heads compared to the ancestry model's attention heads is given by:

$$\frac{LH_a(2d_k + d_v)D}{\sum_1^L c_l(2d_k + d_v)D} = \frac{LH_a}{\sum_1^L c_l} \tag{5}$$

