# OpenReview forum: "Cluster-Learngene: Inheriting Adaptive Clusters for Self-Attention"
_ICLR.cc/2024/Conference — ICLR 2024 Conference Withdrawn Submission_

### Official Review · Reviewer_YtqB · 2023-10-30

**Soundness:** 2 fair
**Presentation:** 4 excellent
**Contribution:** 2 fair
**Rating:** 3
**Confidence:** 5

**Summary:**

The paper meets the senario that we pretrain a model on dataset A (e.g., ImageNet) then adapt it to other datasets (the paper uses other classification datasets such as CIFAR-100). The pretrained model is called the ancestry and the adapted model is called the descendant.

The authors propose to inheirt attention weights from ancestry to the descendant. Only weights from attention weight are inherited. Weights from MLPs are not inherited by default. The main contributions of  the paper are exploring how to inherit the weights, and the main steps are:
1. The attention weights of a attention head is transformed as a point in the space. All points from the ancestry are first clusteted for some centroids. Clusters with many points are assigned with higher priority, while others are with lower priority.
2. The descendant copies centroieds from the step 1 as its initial attention weights. Clusters with higher priority are sampled more frequently than those with lower priority.

In terms of the results, the proposed method perform worse than the most naive method: pretraining the model and fine-tuning the model on the downstream datasets. Further, the experimental settings are strange with many unreasonable assumptions.

**Strengths:**

1. The paper provides the community with sufficient valuable experiences about how to inherit weights from pretrained models, though the results are not satisfying.
2. The paper is well-written and easy to understand.

**Weaknesses:**

1. In algorithm 1, I think different visiting results will lead to different clustering results. For example, if "a" and "b" are two heads in the model, and number of NeighborHds(a) < MinHds and number of NeighborHds(b) > MinHds. We also assume "a" and "b" are neighbors. If we visit "a" first, then "a" will be marked as noise. However, if we visit "first", then "a " will become a member of the cluster initialized by "b". I think a good clustering method for attention weights should have nothing to do with the visit order.
2. The authors pretrain the model on ImageNet and fine-tune it on CIFAR-100, Tiny-ImageNet and other classification datasets. While Tiny-ImageNet is a subset of ImagNet, which will lead to some knowledge leakage, and CIFAR-100 is too simple compared with ImageNet, which makes the results less convising. Also, Results in Table 1, I think, may also indicate some knowledge leakage.
3. In terms of Figure 3 and Figure 4, I think pretraing-fine-tuning, the naive method is the best method perform better than the proposed method.

**Questions:**

1. The subsection "Complexity Analysis" only indicates the initialized complexity of the descendant model. What is the advantage of less model complexity of the initialized attention heads?

---

### Official Review · Reviewer_9WgW · 2023-10-31

**Soundness:** 2 fair
**Presentation:** 2 fair
**Contribution:** 2 fair
**Rating:** 3
**Confidence:** 3

**Summary:**

The manuscript introduces an extension to the existing Learngene technique, proposing an innovative neural network pruning approach. The focus is on clustering the attention heads of a Vision Transformer (ViT) based on their attention scopes. Subsequently, these clusters are ordered in descending sequence, with the centroid of each cluster serving to initialize descendant models. Benchmark tests on ImageNet and some other small-sized datasets reveal the superiority of this approach over certain preceding methods.

**Strengths:**

1. The idea of grouping attention heads with their attending ranges is interesting.
2. The proposed method outperforms some previous methods.

**Weaknesses:**

1. While the idea of clustering attention heads by their attention ranges seems logical sounding, this paper needs a more thorough explanation to back this intuition. Specifically, attention heads with alike average attention ranges but different standard deviations might operate differently. Grouping such heads together may be problematic.
2. This work lacks a clear explanation about the motivation to use the *density-connected* based clustering algorithm. Although clustering by attention weights can be intuitive, it is not clear to me why the *density-connected* heads can be clustered together. For example, assuming the maximal attention range is 5, and there are five attention heads that have attention ranges {1, 2, 3, 4, 5}. If *density-connected* based clustering is used, they should be grouped into a single cluster, which is clearly not correct.
3. The paper lacks a clear explanation about how to determine a cluster's centroid.
4. The comparison is not comprehensive enough. The authors should compare their method with other pruning approaches that are not Learngene-based, i.g. [a].
5. It is not clear if the proposed method can be applied to other visual foundation models, e.g., DINO[b] and SAM[c], and other ViT variants like Swin Transformer [d].
6. What dataset do Fig 3 and Fig 4 use? I guess it is ImageNet, but it should be clearly specified in the captions.
7. This work lacks enough ablation experiments to help readers understand the proposed method. For example, can the clustering algorithm be replaced with other approaches, e.g. K-Means? Given an ancestry, how does the model perform when being applied to different-sized descendant models?
8. It's perplexing why the *pre-training and fine-tuning* lags behind the pruned model in performance in Table 7. A thorough explanation is required since, theoretically, an entirely inherited ancestry model should perform on par with the *pre-training and fine-tuning* model.


---
[a]. Zheng, Chuanyang, et al. "SAViT: Structure-Aware Vision Transformer Pruning via Collaborative Optimization." Advances in Neural Information Processing Systems 35 (2022): 9010-9023.

[b]. Caron, Mathilde, et al. "Emerging properties in self-supervised vision transformers." Proceedings of the IEEE/CVF international conference on computer vision. 2021.

[c]. Kirillov, Alexander, et al. "Segment anything." ICCV 2023.

[d]. Liu, Ze, et al. "Swin transformer: Hierarchical vision transformer using shifted windows." Proceedings of the IEEE/CVF international conference on computer vision. 2021.

**Questions:**

Potential typo:  In section 2.2, it appears there might be a typographical error. Should $D$ actually be $N$?

---

### Official Review · Reviewer_eHH7 · 2023-11-02

**Soundness:** 3 good
**Presentation:** 3 good
**Contribution:** 3 good
**Rating:** 5
**Confidence:** 2

**Summary:**

The authors address the challenge of overgeneralizing large pre-trained models in deep learning, which often overlook task-specific resource constraints. They introduce Cluster-Learngene, a method that condenses knowledge from an ancestry model and initializes descendant models with varying scales of attention heads. The approach adaptively clusters attention heads based on their density characteristics, extracting centroids of attention heads as the "learngene". A priority weight-sharing strategy is introduced to initialize descendant models with varying scales of attention heads. The proposed method aims to strike a balance between accuracy and resource efficiency, especially for deployment on resource-limited devices.

**Strengths:**

Innovative Approach: Cluster-Learngene offers a novel way to condense knowledge from large models, addressing the challenge of overgeneralization.

Efficiency: The method is shown to be more efficient compared to other initialization methods, customizing models according to downstream task resources.

Adaptive Clustering: The adaptive clustering of attention heads based on density characteristics is a unique approach that ensures significant knowledge is preserved.

Priority Weight-Sharing: This strategy allows for the initialization of descendant models with varying scales of attention heads, enhancing flexibility.

**Weaknesses:**

Complexity: The method involves multiple stages, including adaptive clustering and priority weight-sharing, which might increase the complexity of the model initialization process.

Limited to ViTs: The paper primarily focuses on the Vision Transformers (ViTs) architecture. Its applicability to other architectures remains to be seen.

In Equation 4, if T is constant？

How does the priority weight-sharing strategy impact the overall performance of the descendant models?

**Questions:**

Please see above.